# Liver Receptor homolog-1 Regulates Apoptosis of Bovine Ovarian Granulosa Cells by Progestogen Receptor Signaling Pathway

**DOI:** 10.3390/ani12091213

**Published:** 2022-05-08

**Authors:** Dejun Xu, Xiaohan Jiang, Yukun Wang, Shuaifei Song

**Affiliations:** 1Chongqing Key Laboratory of Herbivore Science, College of Animal Science and Technology, Southwest University, Chongqing 400715, China; wyk1014@email.swu.edu.cn (Y.W.); s15236195204@email.swu.edu.cn (S.S.); 2College of Animal Science and Technology, Northwest A&F University, No. 3 Taicheng Road, Xianyang 712100, China; 2015060124@nwafu.edu.cn

**Keywords:** LRH-1, granulosa cells, apoptosis, progestogen

## Abstract

**Simple Summary:**

Liver receptor homolog-1 (LRH-1) is highly observed in tissues with secretory function, such as the ovaries, suggesting that LRH-1 may play an essential role in the ovarian endocrine. In particular, ovarian granulosa cells (GCs) are the functional cells that produce steroid hormones. The fate of GCs directly affects follicular development or atresia. These effects were lost in granulosa-specific LRH-1-knockout mice, showing that LRH-1 is a central modulator of ovarian function. However, the underlying mechanism LRH-1 in the bovine ovaries remains unclear. We aimed to detect the effect of LRH-1 on steroid hormones in bovine GCs and explore the potential relationship between LRH-1 and the fate of GCs. The results show that LRH-1 was specifically highly expressed in GCs of atretic follicles. Mechanistically, LRH-1 induced the apoptosis of bovine GCs by the progestogen receptor signaling pathway. While this finding provided new ideas for the study of follicular atresia, it also provided a theoretical basis for the clinical diagnosis and treatment for infertility in cow.

**Abstract:**

The purpose of the present investigation was to assess the function of LRH-1 on GCs and the mechanisms involved. Here, LRH- was highly expressed in the bovine GCs of atretic follicles. Treatment with 50 μM of LRH-1 agonist (DLPC) significantly induced the expression of LRH-1 (*p* < 0.05). In particular, LRH-1 activation blocked the progestogen receptor signaling pathway via downregulating progesterone production and progestogen receptor levels (*p* < 0.05), but had no effect on 17 beta-estradiol synthesis. Meanwhile, LRH-1 activation promoted the apoptosis of GCs and increased the activity of caspase 3 (*p* < 0.05). Importantly, upregulating the progestogen receptor signaling pathway with progestogen could attenuate the LRH-1-induced proapoptotic effect. Moreover, treatment with progestogen decreased the activity of the proapoptotic gene caspase 3 and increased the expression of antiapoptotic gene Bcl2 in LRH-1 activated GCs (*p* < 0.05). Taken together, these results demonstrate that LRH-1 might be dependent on the progestogen receptor signaling pathway to modulate bovine follicular atresia.

## 1. Introduction

In mammals, the follicle is a basic functional unit of the ovary. However, most follicles will undergo degeneration or atresia, and only a few can develop into mature follicles for ovulation [1,2]. Ovarian granulosa cells (GCs) play a central role in this physiological action through the response to hormone signaling. In particular, the apoptosis of granulosa cells is an induction mechanism of follicular degeneration or atresia [3,4]. Unfortunately, it is still unclear how functional genes expressed in the ovary mediate the apoptosis of granulosa cells. 

Liver receptor homolog-1 (LRH-1), also known as nuclear receptor subfamily 5 group A member 2 (NR5A2), belongs to an orphan nuclear receptor that has been shown to be involved in cholesterol metabolism [5], cell growth [6] and apoptosis [7]. In the ovaries, LRH-1 has been reported to modulate numerous physiological functions, including steroid synthesis [8], pregnancy time course [9], follicles maturation and ovulation [10]. Although it is well known that LRH-1 regulates survival or death in cancer cells [11,12], it remains unknown whether LRH-1 is also involved in the fate of GCs. In particular, additional studies have shown that LRH-1 might regulate granulosa cell steroidogenesis in several animal models [13,14].

Progesterone plays an essential role in ovulation and/or luteal formation through its nuclear receptor [15]. Recently, the progestogen receptor signaling pathway was found to serve a decisive role in controlling cell proliferation and apoptosis [16,17]. However, the reported effects regarding the role of progesterone on apoptosis seem to be conflicting. Some studies have indicated that progesterone can contribute to apoptosis via upregulating the pro-apoptotic protein in cancer cells [18,19]. Conversely, progesterone can also inhibit apoptosis by altering the expression of pro- and antiapoptotic proteins in normal cells [20,21]. Interestingly, several studies report that the progesterone receptor (PGR) signaling pathway decreases apoptotic cell death in bovine [22], human [23] and mouse [24] granulosa cells, suggest that PGR signaling is well established as a survival factor granulosa cell. These findings hint that a redundancy of functions may exist between LRH-1 and PGR signaling in GCs apoptosis. To verify this hypothesis, we investigated the effects of LRH-1 on PGR signaling and apoptosis, and further explored the possible pathway-mediated effects on apoptosis in bovine ovarian granulosa cells.

## 2. Materials and Methods

### 2.1. Chemicals and Reagents 

LRH-1 agonist DLPC (R-2, 3-bis dodecanoyloxy propyl 2-trimethylammonio ethyl phosphate, Cat#: B7661) was purchased from ApexBio (Boston, MA, USA). Rabbit anti-LRH-1 (Cat#: OM108702), rabbit anti-β actin (Cat#: OM241350) polyclonal antibodies were purchased from Omnimabs (Alhambra, CA, USA). Rabbit anti-progesterone receptor (Cat#: 49338) monoclonal antibody was purchased from Signalway antibody LLC (Greenbelt, MD, USA). Rabbit anti-Bcl2 (Cat#: 12789-1-AP) polyclonal antibody was purchased from Proteintech Group, Inc. (Wuhan, China). Rabbit anti-cleaved-caspase 3 (Cat#: ab49822) polyclonal antibody was purchased from Abcam (Cambridge, UK). Unless otherwise indicated, the other reagents and chemicals were purchased from Sigma-Aldrich Chemical Company (St. Louis, MO, USA).

### 2.2. Primary Cell Cultures and Treatment

Bovine ovaries were obtained from a local abattoir (Shaanxi, China) and transported immediately to the laboratory within 6 h with phosphate-buffered saline (PBS) containing penicillin (100 IU/mL) and streptomycin (100 mg/mL) at 27~30 °C. The bovine granulosa cells were isolated from follicles 2 to 8 mm in diameter as described previously [25], and seeded onto 24-well tissue culture plates at a density of 1 × 10^6^/mL of serum-free medium. The serum-free medium was composed of DMEM/F12 (Gibco, Invitrogen Life Technologies), 0.1% bovine serum albumin (BSA) (*w*/*v*), sodium bicarbonate (10 mM), sodium selenite (4 ng/mL), transferrin (2.5 mg/mL), epidermal growth factor (EGF) (5 ng/mL), bovine insulin (10 ng/mL), non-essential amino acid mix (1×) and penicillin–streptomycin liquid (1×). Then, granulosa cells were cultured in an atmosphere of 5% CO_2_ at 37 °C. DLPC was diluted with ethanol, and, in all of the experiments, the concentration of ethanol was diluted to below 0.1% (*w*/*v*). After culturing for 24 h, the bovine granulosa cells were treated with various concentrations of DLPC (50, 100, 150, 200 μM) or/and progestogen (5, 10, 100 μM) for 24 h, respectively.

### 2.3. Steroid Assay

After the treatment with DLPC for 24 h, the cultured medium of granulosa cells was collected to measure 17 beta-estradiol and progesterone using competitive ELISA Kit (Cat#: 582251, Cat#: 582601; Cayman, Ann Arbor, MI, USA). The ELISA procedure was performed according to Cayman’s ELISA kit instructions, and the optical density (OD) values at 420 nM were measured by a microplate reader (Thermo, Waltham, MA, USA). An eight-point standard curve was obtained for each series of analyses. Finally, the concentrations of 17 beta-estradiol and progesterone were calculated using the formula derived from the standard curve in cultured medium. 

### 2.4. Measurement of Cell Viability

The viability of granulosa cells was determined using a Cell Counting Kit-8 (Beyotime, Hangzhou, China). According to the kit instruction, granulosa cells were incubated for 1 h with 10 μL of CCK-8 solutions at 37 °C in 100 μL of culture medium. Then, the optical density (OD) values of cell viability at 450 nm were measured by a microplate reader (Thermo, Waltham, MA, USA).

### 2.5. Apoptosis Measurements 

Annexin V-fluorescein isothiocyanate (FITC) staining reagent (Vazyme, China) was used to measure the apoptosis of granulosa cells. Briefly, the cells were washed three times with cold PBS by centrifugation at 300 g for 5 min. The cell pellet was resuspended in 100 μL of binding buffer containing 5 μL of Annexin V-FITC and 5 μL of PI staining solution at a density of 1 × 10^6^/mL, and incubated for 10 min at room temperature in the dark. Then, 400 μL of binding buffer was added to the stained cell suspension. The stained cells were detected within 1 h by a flow cytometry instrument (Beckman Coulter, Brea, CA, USA).

### 2.6. Immunohistochemistry

After fixing with 4% paraformaldehyde (Beyotime, Hangzhou, CA, China) overnight at 4 °C, the bovine ovaries were embedded with paraffin and cut into 5 µm sections for immunostaining. The ovary sections were dewaxed and dehydrated with sodium citrate buffer (10 mM sodium citrate, 0.05% Tween 20, pH = 6) for antigen retrieval. Then, the sections were blocked for 1 h at room temperature in QuickBlock™ blocking buffer (Beyotime, Hangzhou, China) and incubated with anti-LRH-1 antibody (1:200) or non-immune rabbit IgG (1:200, for negative controls) overnight at 4 °C, respectively. Incubated antibodies were diluted with TBST (20 mM Tris-HCl, 150 mM NaCl, 0.05% Tween 20) containing 5% (*w*/*v*) non-fat dry milk. After washing three times for 5 min each time with TBST, the sections were incubated for 1 h at room temperature with anti-rabbit secondary antibody (1:5000; Beyotime, China). The samples were stained for 10 min with SignalStain^®^ Boost IHC Detection Reagent (Cell signaling, Danvers, MA, USA). After washing, the sections were counterstained for 20 s with haematoxylin. Finally, the stained section images were captured by a digital microscope (Nikon, Tokyo, Japan). The mean density of LRH-1 was analyzed with Image J software (National Institutes of Health, Bethesda, MD, USA).

### 2.7. Quantitative Real-Time PCR

After treatments, granulosa cells were harvested for total RNA extraction using Trizol reagent (Takara, Kusatsu, Japan). After quantifying with a spectrophotometer (Thermo, Waltham, MA, USA), the first-strand cDNA was synthesized using 5 × All-In-One RT MasterMix (Abm, Vancouver, Canada). The targeted cDNA was quantified by CFX96TM Real-Time PCR (Bio-Rad, Hercules, CA, USA) using the EvaGreen 2 × qPCR MasterMix-no Dye (Abm, Vancouver, Canada). The bovine-specific primer sequences for target genes are listed in Table 1. The β-actin was used as a reference gene. The quantitative real-time PCR protocol was as follows: 30 s at 95 °C; 40 cycles of 5 s at 95 °C; and 30 s at 60 °C. Relative expression of target genes was calculated using the 2^△△Ct^ method.

### 2.8. Western Blotting

Granulosa cells were lysed with RIPA buffer (Beyotime, Hangzhou, China) on ice. Then, the total proteins were harvested by centrifugation at 12,000× *g*, for 10 min at 4 °C, and quantified using a BCA protein assay kit (Beyotime, Hangzhou, China). The proteins were mixed with SDS–PAGE sample loading buffer (Beyotime, Hangzhou, China) and boiled for 5 min at 100 °C. Approximately 20 μg of total proteins were submitted to each gel electrophoresis and separated by 15% SDS-PAGE. After transferring onto nitrocellulose membranes, the proteins were blocked in TBST containing 5% (*w*/*v*) non-fat dry milk with slight shaking for 1 h at room temperature and incubated overnight at 4 °C with anti-LRH-1 (1:1000), anti-Bcl2 (1:500), anti-cleaved-caspase 3 (1:500) and anti-β-actin (1:1000), respectively. Incubated antibodies were diluted with TBST containing 5% (*w*/*v*) non-fat dry milk. After washing with TBST solution, the proteins were incubated for 1 h by goat anti-rabbit IgG(H+L)-HRP (1:5000; Sungene Biotech, Tianjin, China) with slight shaking at room temperature. After washing at least four times, the protein bands were exposed to X-ray film for visualization with ECL (Millipore, Burlington, VT, USA). Then, the band intensities were measured using Quantity One software (BioRad, Hercules, CA, USA).

### 2.9. Immunofluorescence

After treatment, granulosa cells were fixed for 30 min with 4% paraformaldehyde and then washed three times with PBS. The fixed cells were permeabilized with 0.2% Triton X-100 (Beyotime, Hangzhou, China). After blocking with QuickBlock™ blocking buffer, the cells were incubated with anti-progesterone receptor (1:200) overnight at 4 °C. After being washed three times, the cells were incubated with Alexa Fluor 594-labelled goat anti-mouse IgG (Proteintech, Wuhan, China) for 1 h with slight shaking at room temperature. Then, the cells were counterstained with 4,6-diamidino-2-phenylindole (DAPI), and imaged using a fluorescence microscope (Nikon, Tokyo, Japan). The fluorescence intensity of progesterone receptor was analyzed with Image J software (National Institutes of Health, Bethesda, MD, USA).

### 2.10. Statistical Analysis

All experiments were repeated at least three times unless specified otherwise. Data were shown as mean ± standard error of the mean (SEM). All statistical data were analyzed with SPSS 20.0 statistical software (SPSS, Chicago, IL, USA). One-way analysis of variance (ANOVA) followed by Duncan’s test was performed to compare means between multiple groups, whereas statistical difference with two groups was analyzed by the student’s *t*-test. A *p* value < 0.05 was considered as a statistically significant difference.

## 3. Results

### 3.1. LRH-1 High Expression in the Atretic Follicles

By performing immunohistochemistry assays, the results showed that LRH-1 was mainly present in the atretic follicles, but not the healthy follicles (Figure 1A). Further observation showed that LRH-1 was strongly presented in the GCs of atretic follicles (Figure 1A). Compared with the healthy follicles, more LRH-1 was observed in the GCs of atretic follicles (*p* < 0.01, Figure 1B), showing that LRH-1 might regulate follicular atresia.

### 3.2. DLPC Induced LRH-1 Expression

To upregulate the activity of LRH-1, GCs were treated with various concentrations of LRH-1 agonist (DLPC). As shown in Figure 2A, the mRNA levels of LRH-1 were significantly increased by treatment with 50 μM DLPC (*p* < 0.05). By performing Western blotting analysis, the result showed that the protein abundance of LRH-1 was also increased by treatment with 50 μM DLPC (*p* < 0.001; Figure 2B,C). However, the high concentration of DLPC (100, 150 μM) had no effect on the LRH-1 levels. The results suggested that LRH-1 is efficiently induced by treatment with 50 μM DLPC. 

### 3.3. LRH-1 Activation Blocked Progestogen Receptor Signaling Pathway

To investigate the role of LRH-1 on steroid hormone synthesis in primary granulosa cells, the concentrations of 17 beta-estradiol (E2) and progesterone (P4) were measured by ELISA assay after treatment with different concentrations of DLPC for 24 h. Although previous work showed that 50 μM DLPC increased the protein levels of LRH-1, treatment with 50 or 100 μM DLPC had no effect on E2 synthesis (*p* > 0.05, Figure 3A), whereas the progesterone synthesis was blocked by DLPC in a concentration-dependent manner (*p* < 0.05, Figure 3B). In addition, the mRNA levels of the progesterone receptor (PGR) were decreased by treatment with DLPC (*p* < 0.05, Figure 3C). By performing immunofluorescence staining, it was revealed that the protein level of PGR was decreased by treatment with 50 μM DLPC (*p* < 0.05; Figure 3D,E). Interestingly, the inhibition effect of DLPC on PGR was abolished by the addition of 10 μM P4 (*p* < 0.001; Figure 3D,E). These results suggested that LRH-1 inhibits the progestogen receptor signaling pathway, but not 17 beta-estradiol signaling in bovine GCs.

### 3.4. LRH-1 Mediated Apoptosis via Progestogen Signaling

Compared with the control group, a significant decrease in cell viability was observed in the treatment with DLPC groups (*p* < 0.01, Figure 4A). To investigate how LRH-1 regulated the cell survival, the bovine GCs were exposed with 50 μM of DLPC or/and various concentrations of P4 for 24 h. As shown in Figure 4B,C, LRH-1 activation with DLPC induced the apoptosis of GCs. Although DLPC had no effect on Bcl-2, the level of cleaved-caspase 3 was up-regulated by treatment with DLPC (*p* < 0.05, Figure 4D–F). Interestingly, treatment with P4 at concentrations of 5 or 10 μM could attenuate the DLPC-induced apoptosis (*p* < 0.05, Figure 4B,C), whereas a high concentration of P4 (100 μM) had little effect on DLPC-induced apoptosis (*p* > 0.05, Figure 4B,C). Western blotting showed that the protein level of Bcl-2 was increased by treatment with P4 at various concentrations, except for 5 μM in DLPC-exposed GCs (*p* < 0.05, Figure 4D,E). In addition, treatment with P4 significantly decreased the level of cleaved-caspase 3 in DLPC-exposed GCs, and the 10 μM P4 group had the lowest level of cleaved-caspase 3 (*p* < 0.05, Figure 4D,F). Taken together, these findings reveal that progestogen signaling plays an important role in LRH-1 induced apoptosis in bovine GCs.

## 4. Discussion

The apoptosis of granulosa cells is a mechanism underlying ovarian follicle atresia [2]. Previous studies indicate that steroid hormone signaling (e.g., 17 beta-estradiol, progesterone, androgen) might modulate the apoptosis of granulosa cells [26,27]. LRH-1 has been shown to modulate the expression of steroidogenic enzymes such as P450scc [28], 3β-hydroxysteroid dehydrogenase [29] and CYP17 [30], suggesting that LRH-1 might serve an important role in ovarian steroidogenesis. It would thus be of interest to test whether LRH-1 is involved in ovarian follicle atresia through steroid hormone signaling.

LRH-1 was expressed in numerous organs, including the liver, pancreas colon and ovaries [31]. Here, we also found that LRH-1 was present in bovine ovaries, and was strongly expressed in the granulosa cells of atretic follicles. Saxena demonstrated that LRH-1 cannot stimulated the estrogen or progesterone production of granulosa cells in the absence of FSH, but significantly amplified the stimulatory effects of FSH on progesterone production [8,13]. The function of LRH-1 on ovarian steroidogenesis has been further explored in the LRH-1^+/−^ mouse model. These LRH-1-knockout mice exhibited a normal ovarian exocrine function and a normal oestradiol synthesis, whereas a decrease in progesterone production was observed in response to gonadotropin administration in these mice [14]. Consistent with Labelle-Dumais’ observers [14], our data showed that LRH-1 activation had no effect on oestradiol production. However, we found that LRH-1 activation inhibited progesterone production in granulosa cells cultured in vitro. One possible explanation for these quantitative differences is that the hormone-stimulated progesterone biosynthetic pathway in vivo and vitro is different or even opposite. Traditionally, LRH-1 was thought to be a promoting factor for steroid synthesis [13,32]. Interestingly, our results provide direct evidence that disrupting the normal activity of LRH-1 could also inhibit progesterone production. Progesterone is thought to mediate its function via the nuclear progesterone receptor [33]. Further analysis revealed that LRH-1 activation decreased the transcription and translation levels of the progesterone receptor. In addition, treatment with progesterone could abolish the effect of LRH-1 activation on the progesterone receptor, suggesting that LRH-1 mediates the progestogen receptor signaling pathway in granulosa cells cultured in vitro. 

Accumulate evidence demonstrates that progestogen receptor signaling has an antiapoptotic role in the regulation of granulosa cell survival in most species studied [22,23,34,35]. The presence of the LRH-1-mediated progestogen receptor signaling pathway in granulosa cells raises the question as to whether there may be regulation actions of LRH-1 on follicle atresia. As expected, our data support the notion that LRH-1 activation induced apoptosis in granulosa cells cultured in vitro. In support of this, LRH-1 activation upregulated the activity of the proapoptotic gene caspase 3. However, LRH-1 has been thought to promote tumor cell growth via an increased synthesis of local 17 beta-estradiol in cancer cells [36,37]. In this study, LRH-1 activation had almost no effect on 17 beta-estradiol produce, suggesting that LRH-1 does not depend on 17 beta-estradiol signaling to affect granulosa cells’ survival. Interestingly, treatment with progestogen could attenuate the LRH-1-induced proapoptotic effect in this study. Furthermore, progestogen increased the expression of antiapoptotic gene Bcl2 and decreased the activity of the proapoptotic gene caspase 3 in LRH-1-activated granulosa cells. There observations demonstrate that LRH-1 might mediate apoptosis through progestogen signaling in bovine granulosa cells.

## 5. Conclusions

In summary, treatment with LRH-1 agonist (DLPC) inhibited progestogen production and progestogen receptor expression. In addition, LRH-1 activation with DLPC induced the apoptosis of granulosa cells and upregulated the activity of caspase 3. Importantly, progestogen could attenuate the LRH-1-induced proapoptotic effect by upregulating Bcl2 and downregulating cleaved-caspase 3. The present study provides the first direct evidence that LRH-1 activation plays a proapoptotic role in granulosa cells cultured in vitro via downregulating the progestogen receptor signaling pathway.

## Figures and Tables

**Figure 1 animals-12-01213-f001:**
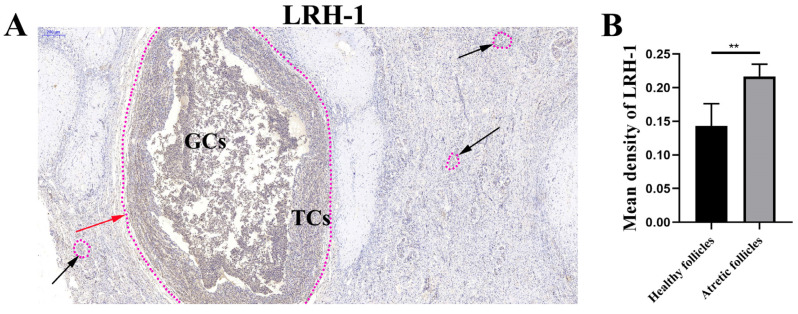
LRH-1 high expression in the atretic follicles. (**A**) Immunochemistry assay was performed to visualize the localization of LRH-1 in bovine ovarian cells. Immuno-specific staining was brown, indicating LRH-1-positive cells. Immunohistochemistry was performed on three different ovarian slides from each of three bovines. Atretic follicles were indicated with the red arrow, whereas healthy follicles were indicated with black arrows. Bar: 200 μm. (**B**) The LRH-1 levels of granulosa cells in atretic follicles or healthy follicles. Data are shown as the means ± SEM of three biological replicates. ** *p* < 0.01, comparing the indicated groups. TCs, theca cells; GCs, granulosa cells.

**Figure 2 animals-12-01213-f002:**
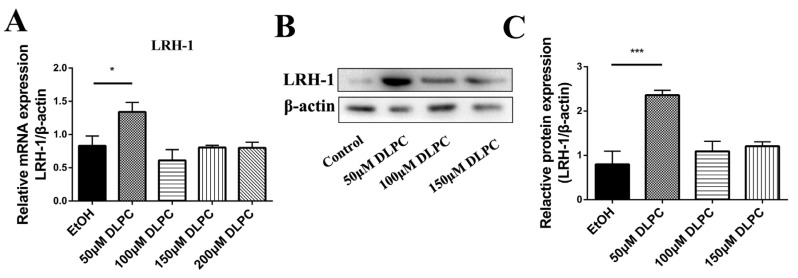
DLPC induced LRH-1 expression. (**A**) The mRNA levels of LRH-1 in granulosa cells by treatment with various concentrations of LRH-1 agonist (DLPC). (**B**) Western blotting of LRH-1 protein in granulosa cells by treatment with various concentrations of DLPC. (**C**) The protein levels of LRH-1 in the different groups. Data are expressed as the mean ± SEM from three biological replicates. * *p* < 0.05, *** *p* < 0.001, comparing the indicated groups.

**Figure 3 animals-12-01213-f003:**
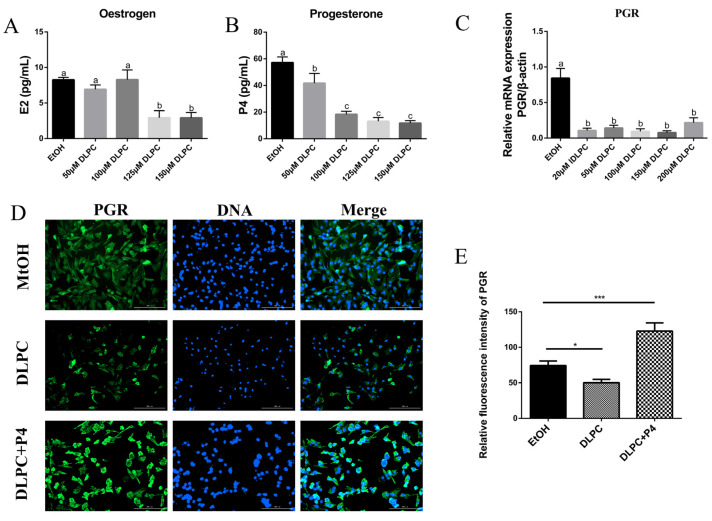
DLPC blocked progestogen receptor signaling pathway. (**A**) The effect of DLPC on 17 beta-estradiol (E2) production of granulosa cells. The concentration of oestradiol was detected by ELISA assay. (**B**) The effect of DLPC on progestogen (P4) production of granulosa cells. (**C**) The mRNA levels of progestogen receptor in granulosa cells by treatment with DLPC. (**D**) Immunofluorescence of progestogen receptor in granulosa cells by treatment with 50 μM DLPC or/and 10 μM P4. (**E**) Quantification of progestogen receptor was analyzed with Image J software. Bar: 200 μm. Data are expressed as the mean ± SEM from three independent replicates. Values with different letters (a, b, c) indicate significant differences in bars (*p* < 0.05). * *p* < 0.05, *** *p* < 0.001, comparing the indicated groups.

**Figure 4 animals-12-01213-f004:**
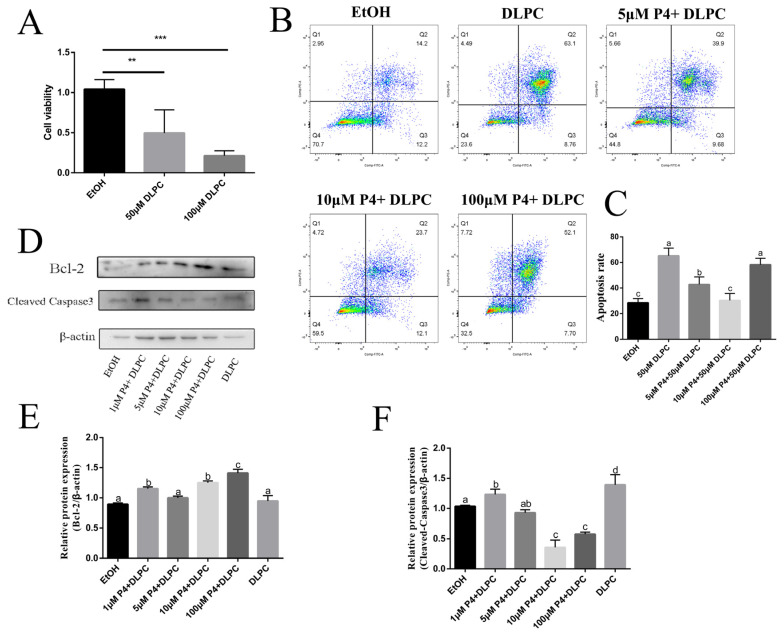
DLPC activation promoted apoptosis via progestogen signaling. (**A**) The viability of granulosa cells was shown by CCK-8 assay in different treatments. (**B**) Representative images of apoptosis with annexin V-FITC staining by a flow cytometry instrument in 50 μM DLPC-, 50 μM DLPC + 5 μM P4-, 50 μM DLPC + 10 μM P4-, 50 μM DLPC + 100 μM P4-exposed granulosa cells. (**C**) The percentage of apoptosis in the different treatment groups. (**D**) Western blotting of Bcl2 and cleaved-caspase 3 in granulosa cells by treatment with 50 μM DLPC, 50 μM DLPC + 1 μM P4, 50 μM DLPC + 5 μM P4, 50 μM DLPC + 10 μM P4 and 50 μM DLPC + 100 μM P4, respectively. (**E**,**F**) The ratios of Bcl-2 to β-actin, cleaved-caspase 3 to β-actin expression were normalized, and the values were shown, respectively. Data are shown as the means ± SEM of three biological replicates. Values with different letters (a, b, c) indicate significant differences in bars (*p* < 0.05). ** *p* < 0.01, *** *p* < 0.001, comparing the indicated groups.

**Table 1 animals-12-01213-t001:** Sequences for primers used in quantitative real-time RT-PCR.

Gene Name	Primer Sequences (5′–3′)	GenBank Accession No.
LRH-1	F: TCTTTGAACACCACCCAATACCA	XM_015470800.1
	R: ATCTGCTGGTCGGAAAGGC	
PGR	F: TCCCCCCACTGATCAACTTG	NM_001205356.1
	R: TCCGAAAACCTGGCAGTGA	
β-actin	F: TCACCAACTGGGACGACA	NM_173979.3
	R: GCATACAGGGACAGCACA	

## Data Availability

The data that support the findings of this study are available on the request from the corresponding author.

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
