# Peer review of "Liver Receptor homolog-1 Regulates Apoptosis of Bovine Ovarian Granulosa Cells by Progestogen Receptor Signaling Pathway"

_animals, 2022, doi:10.3390/ani12091213_

Round 1
Reviewer 1 Report
Dear Authors,
The manuscript entitled “Liver receptor homolog-1 regulates apoptosis of bovine ovarian granulosa cells by progestogen receptor signaling pathway” has been revised. The study aimed to investigate the effects of LRH-1 on PGR signaling and apoptosis, and further explored the possible pathways mediating effects on apoptosis in bovine ovarian granulosa cells. The manuscript is scientifically sound. The hypothesis was adequately tested with the methodology applied. The results clearly show when LRH-1 is activated, it has a proapoptotic role in in vitro granulosa cells. The information is relevant to the research field. Only minor issues, such as grammar and typos, were highlighted in the file attached. Please consider the comments in the file to improve the manuscript. Congratulations to the authors for the very interesting work.

Author Response
We appreciate your invaluable suggestion and comment. The grammar and typos had been corrected.

Reviewer 2 Report
The papres is interesing and is well written, However I havee some suggestions for the authors:
Please give the experimental number of different experiments (n, biological and replicates)
Please give the solutions used for washing steps and antibody incubation for the western blot and immunohistochemestry
Figure 2B, the western blot shows that the standard protein (actin) is of less intensity than in the other lines. I wonder this could be why author found an increase in LHR-1. Same in Fig 4D, where in the last line seems that actin has a decrease intensity than in the other lines.
Fig 3. Please comment regarding the results that oestrogen and progesterone levels decrease with higher concentrations of DPLC
Authors state that "he bovine GCs were exposed to various concentrations of P4 after pretreatment with 50 μM DLPC for 24 hr". If P4 was added after DLPC, which induce apoptosis How is possible that P4 could have prevent this effect?
I think authors could include some results regarding the mechanism, (e.g.) using a P4 receptor inhibitor, or make a kockdown of this protein.
Author Response
Please give the experimental number of different experiments (n, biological and replicates)
Response: We appreciate the reviewer’s invaluable suggestion and comment. The experimental number of different experiments was given in the revised manuscript.
Please give the solutions used for washing steps and antibody incubation for the western blot and immunohistochemestry
Response: We appreciate the reviewer’s invaluable suggestion. The western blot and immunohistochemestry protocols were given in manuscript.
Figure 2B, the western blot shows that the standard protein (actin) is of less intensity than in the other lines. I wonder this could be why author found an increase in LHR-1. Same in Fig 4D, where in the last line seems that actin has a decrease intensity than in the other lines.
Response: Due to the shorter exposure time of the actin bands in Figure 4D, the intensity appears weaker than that actin in Figure 2B. It doesn't affect the results, the gels in different figures were normalized by the corresponding standard protein (actin).
Fig 3. Please comment regarding the results that oestrogen and progesterone levels decrease with higher concentrations of DPLC
Response: In this study, the cultured medium of granulosa cells was collected to measure oestrogen and progesterone. The higher concentrations of DPLC induced apoptosis of granulosa cells, the production and secretion of steroids were blocked caused by apoptosis in granulosa cells.
Authors state that "the bovine GCs were exposed to various concentrations of P4 after pretreatment with 50 μM DLPC for 24 hr". If P4 was added after DLPC, which induce apoptosis How is possible that P4 could have prevent this effect?
Response: We appreciate the reviewer’s invaluable comment. We are sorry that this description is a clerical error. The correct protocols of treatments had been given in the original materials and methods state that “After cultured 24 hr, the bovine granulosa cells were treated with various concentrations of DLPC or/and progestogen (5, 10, 100 μM) for 24 h, respectively”. The incorrect description has been corrected in the manuscript.
I think authors could include some results regarding the mechanism, (e.g.) using a P4 receptor inhibitor, or make a kockdown of this protein.
Response: We appreciate the reviewer’s invaluable suggestion. In this study, progesterone has beneficial effect on apoptosis caused by DLPC, and the relationships among DLPC, progesterone and its receptors are clear. It is well known that progesterone plays biological functions through its receptors. These results positively demonstrated that P4 regulates DLPC-induced apoptosis through its receptor. As reviewer’s suggestion, it is an interest work to test the effect of P4 inhibition on these results. However, it is unfortunately difficult to carry out such an experiment in this study because of the experimental conditions.

Reviewer 3 Report
The manuscrit by Xu et al. is well designed and well written. The hypostesis is properly formulated. The methodology and results description are adequate. However there are some minor corrections that should be addressed.
- line 39: should be "In mammals"
- could you provide ethicalcommittee agreement if needed
- please unify p value (written lowercase or upercase)
- In Figure 4 description you used lowercase (line 250) instead of upercase
Author Response
line 39: should be "In mammals"
Response: We appreciate the reviewer’s invaluable suggestion and comment. The “mammalian” had been revised.
could you provide ethical committee agreement if needed
Response: We appreciate the reviewer’s invaluable comment. Although this research investigated the function of liver receptor homolog-1 on bovine granulosa cells, there were no treatment and experimentation with live bovine at all. Also, there are no animal experiments were used in this study. The ovaries used in this research were collected from bovine that was original death at a commercial slaughterhouse. This study involved experiments on ovarian granulosa cells, not experimental animals. Slaughter and animal treatment of bovine were not involved in this experiment, as the samples were purchased from livestock products of slaughterhouse. Assuredly, this study is not applicable ethical committee agreement, as only bovine found dead were used.
please unify p value (written lowercase or upercase)
Response: We appreciate the reviewer’s invaluable suggestion. p value had been unified.
In Figure 4 description you used lowercase (line 250) instead of uppercase
Response: We appreciate the reviewer’s invaluable suggestion. They had been revised.

Round 2
Reviewer 2 Report
I do not have more questions.